# Generational mutation patterns in a honey bee Deformed wing virus via infectious clones

Anthony Nearman [1*], Alriana Buller-Jarrett [1,2], Dawn Boncristiani[1], Eugene Ryabov [3], Yanping Chen [1], Jay D. Evans[1]

1 USDA-ARS Bee Research Lab, Maryland, United States of America, 2 Oxon Hill High School, Maryland, United States of America, 3 Department of Cell and Molecular Sciences, The James Hutton Institute, Scotland, United Kingdom

* anthony.j.nearman@gmail.com

## Abstract

Efforts to improve honey bee colony health continue due to persistent high loss rates. A major focus in this area is Deformed wing virus (DWV), a key driver of colony loss. The application of modern molecular techniques has characterized the DWV genome and its high mutational rate that enables the formation of diverse quasi-species populations capable of evading host immune responses, while other work has led to the development of DWV clones suitable for sequence-specific tracking of viral dynamics. In this work we combine knowledge of these efforts to track the mutational progression in a DWV clone surrounding an area of low nucleotide diversity and compare it to its wild-type source. We achieve this through amplicon sequencing of the structural viral protein, VP2, after incubation across three generations and multiple host genetic sources. Inocula were injected into pupae, allowed to replicate, then extracted for a further two generations of injections. For the final injection generation, recipient pupae were injected with preparations from either the same genetic source or cross-fostered from other colonies. Overall, we compared the mean number and type of mutations, their proportional abundance in the read pool, and specific locations across strains. Sequencing results indicate a limited number of mutational hotspots, which were driven by silent mutations in the final injection generation of the wild-type strains. No significant differences were found among other mutation types, cross-fostering status, or interactions with host genetics. This work is an initial attempt at examining viral dynamics in a cloned system across multiple generations and treatment groups. The results provide valuable insights, which may further enhance our understanding of viral dynamics and potentially improve future honey bee therapeutics.

**Data availability statement:** All metadata and summary analysis from sequence data is available as supplemental information (S1 Table). All sequence data is available at NCBI under BioProject ID: PRJNA1295475.

**Funding:** The author(s) received no specific funding for this work.

**Competing interests:** The authors have declared that no competing interests exist.

## Introduction

Managed honey bee colonies in the US continue to experience high loss rates [1,2], due to a combination of factors, including pathogens, parasites, poor nutrition, and pesticide exposure – each acting alone or in synergy [3–6]. Among the various contributing factors, Deformed wing virus (DWV) is of particular concern due to its association with colony loss [7–9] and increased virulence when vectored by the ectoparasitic mite, *Varroa destructor* [7,10,11]. The need for new knowledge and potential solutions is great, as multiple strains of DWV have been detected worldwide [12–15].

Advances in next-generation sequencing technology have begun to reveal key insights into viral molecular physiology. Portions of the DWV genome display high polymorphic diversity, which may generate quasi-species capable of evading host immune responses [16]. On the other hand, identifying regions of low diversity may reveal immutable amino acid residues involved with cellular entry or other key sites for infectivity. Recent developments allow the successful creation and implementation of viral clones based on the DWV genome [17,18]. While these clones were motivated by efforts to encode reporter proteins for visualization, they can also be used to study mutational processes within a known population. This information could then be used to identify the effect of host variability on viral replication, thus increasing the resolution of previous studies characterizing variable host immune responses to pathogens [19–23] and studies of viral evolution under different modes of transmission [24]. Further, while RNA interference shows promise as an antiviral control strategy for bees [25,26] and indeed is a key endogenous immune response [27], studies exploiting cloned variants of DWV show that RNAi is less effective against infection when targeting regions with high sequence diversity [16]. Locating regions in the DWV genome that differ in mutation rate can lead to more effective controls that avoid hotspots for diversity.

Here we examined the distribution, abundance, and type of mutations accumulated by both a clone of DWV (NanoLuc) strain [18] and wild-type source populations (Wild Type) across multiple generations and within or between multiple colonies. We hypothesized that mutations would accumulate over successive generations and occur in similar genomic locations, regardless of viral source. Additionally, we anticipated that host genetic variation would influence mutation patterns. To test these hypotheses, we designed pairs of primers for clone-specific and non-specific amplicons encompassing a 433 nucleotide region of the gene encoding for the structural viral protein, VP2 (Fig 1). This region is directly adjacent to the clone insert, making it ideal for primers either spanning the insert and VP2 or the wild-type Leader Protein and VP2, thus selecting for each strain. We then produced amplicons before and after infecting hosts for three generations. The results suggest a generational accumulation of silent mutations in genomic locations that are similar across clone- and wild-type derived quasi-species, while no host effects were detected. This study represents the first attempt at examining location-specific mutations in a cloned honey bee virus under controlled experimental conditions.

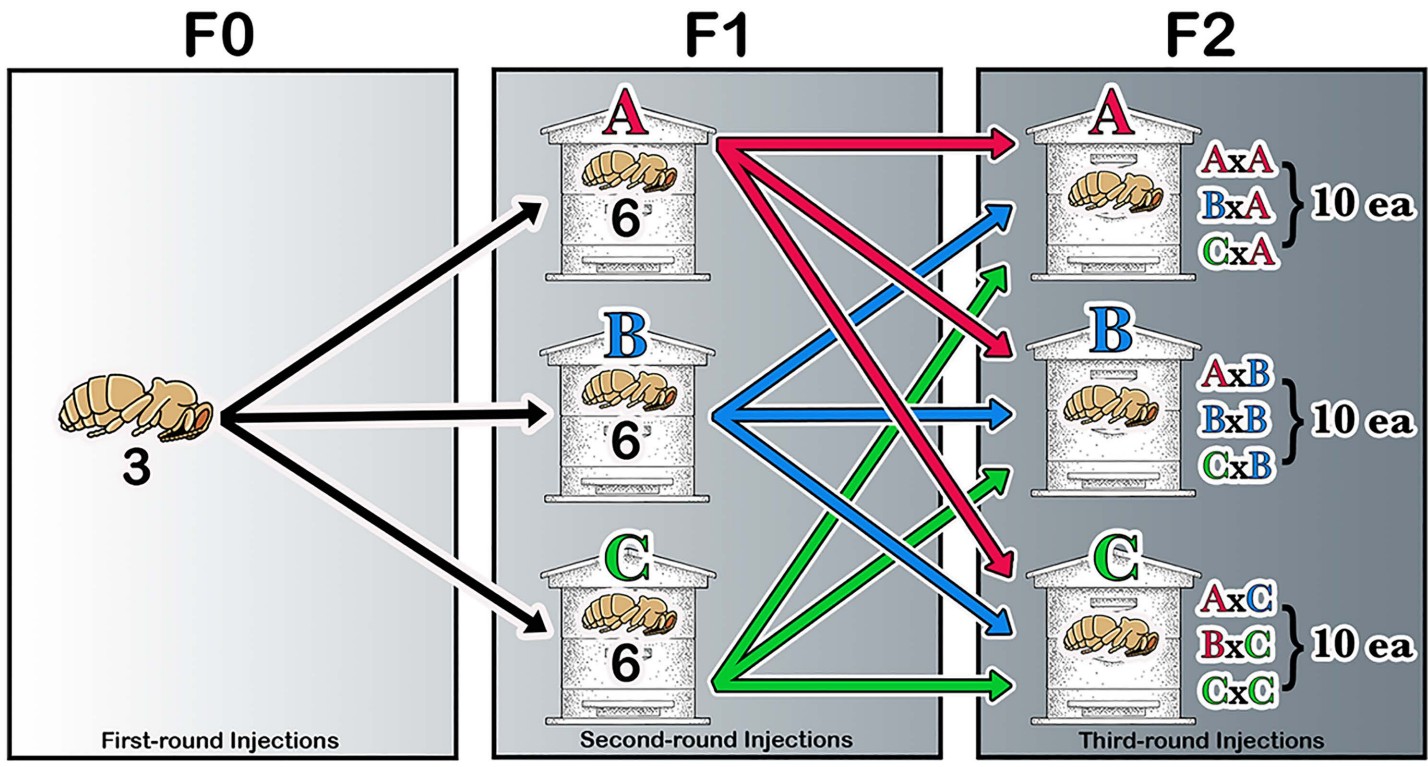

**Arrows = Filtration & Innoculation**

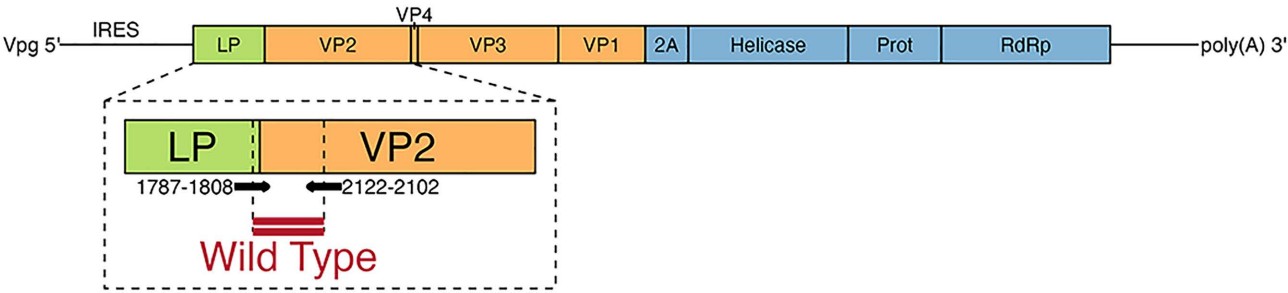

a) Wild Type DWV (AJ489744)

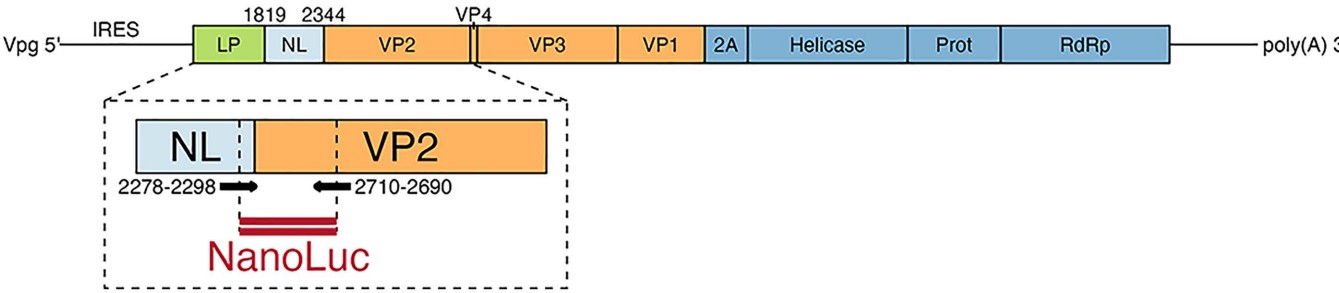

b) DWV-NanoLuc Clone (MW748703)

**Fig 1. Schematic of injection schedule (top) and primer and sequenced read locations (bottom) for a) wild-type DWV (AJ489744) and b) DWV derived NanoLuc Clone (MW748703).** RNA extractions were prepared and sequenced for each injection round as described.

## Results

*Recipient colony affects viral load, but there is no interaction effect.* In consecutive trials viral load did not differ significantly among colony recipients in rounds 1 and 2 (Fig 2). Wild-type virus was present at higher copy numbers in both the initial inoculum and in recipients, as expected. Colony A showed higher viral loads than did Colonies B and C across all inocula in round 3, and there was no significant interaction effect whereby certain pairings resulted in unusually high or low levels of infection in round 3 (Supplemental S1 Fig).

*Positional summary suggests region-specific accumulation of mutation types across injection generations.* Regardless of strain, early sequence base variations (bases 150–200) are more prone to missense mutations than silent mutations but rarely accumulate above 1% proportional representation. By the final injection generation, silent mutations have site-biased accumulation in the wild-type strain at proportions greater than 1% representation. Overall, the wild-type strain indicates a greater accumulation of quasi-species, though similar site-specific patterns were observed in both strains (Fig 3).

*Wild-type strains accumulated a greater number of unique mutations driven by silent mutations.* Across all samples, wild type strains accumulated a normalized average of 15.2 +/- 0.8 mutations compared to the NanoLuc cloned strain with 12.2 +/- 0.5 (W = 28, p < 0.004). This appears to be due to the mean number of silent mutations (wild type 7.4 +/- 1.0, NanoLuc 3.0 +/- 0.3, W = 4, p < 0.001), whereas they did not differ in their number of missense mutations (wild type 7.8 +/- 0.6, NanoLuc 9.1 +/- 0.5). Comparison of nonsense mutations was not possible due to low counts after normalization (Fig 4).

*Wild-type strains showed a greater proportion of reads with silent mutations in the F2 generation.* Similar to the number of unique mutations, their proportional representation in the read pool was also greater among wild-type

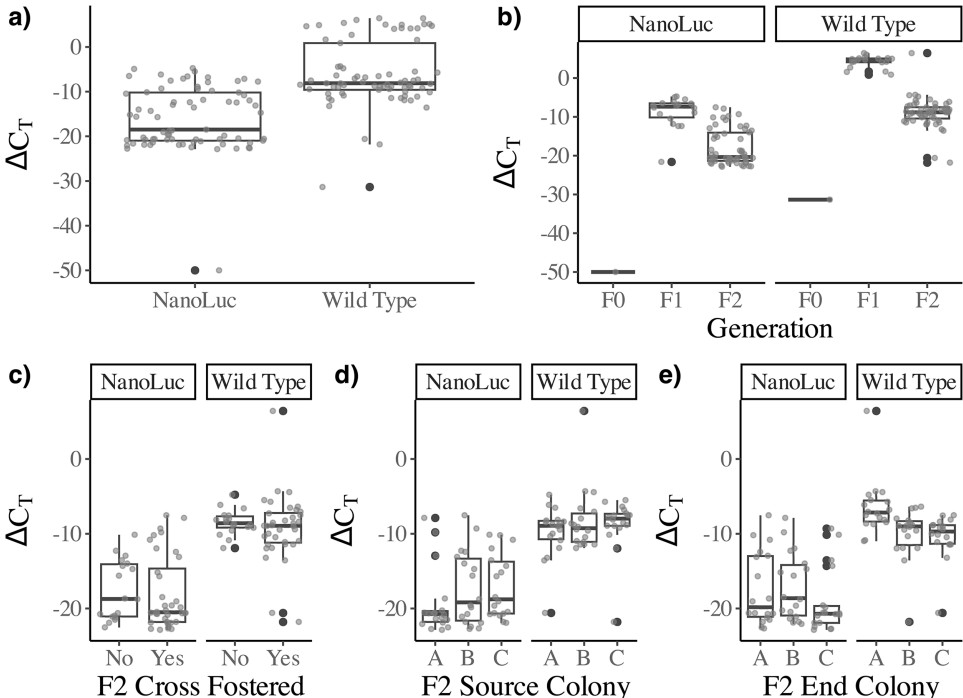

**Fig 2. Relative transcript abundance (scaled to honey bee transcripts for β-actin) for NanoLuc clone-specific and wild-type primers a) globally; b) across successive injection generation; c) final generation cross-fostering status; d) final generation source colony; e) final generation end colony.**

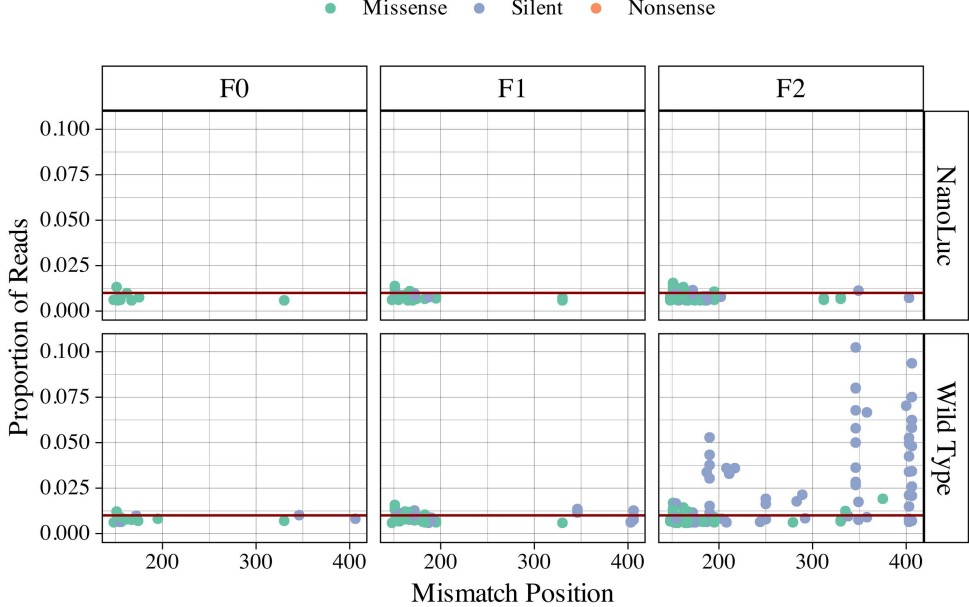

**Fig 3. Position and normalized proportional representation of mutation types found in the DWV structural viral protein, VP2, for the overlapping regions of the NanoLuc cloned virus and wild type strains across successive injection generations (F0 n = 1, F1 n = 3, F2 n = 6) for all samples.** Horizontal line identifies 1% proportional representation.

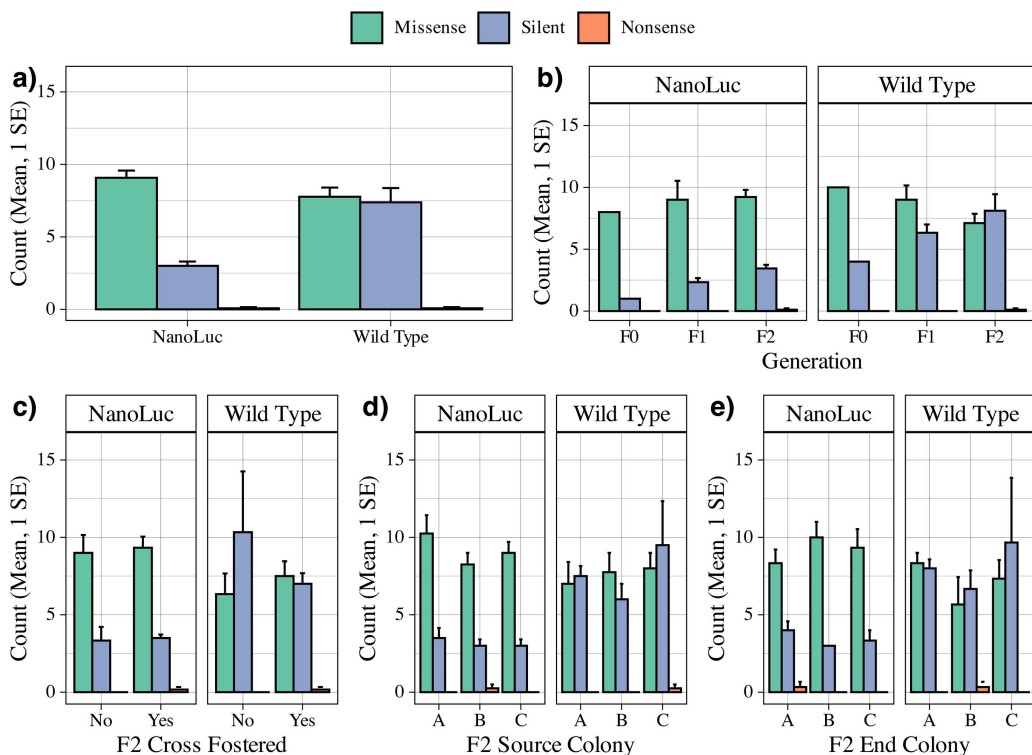

**Fig 4. Comparison of the normalized count mean and 1 SE of mutation types across isolated DWV strains a) globally; b) across successive injection generation; c) final generation cross fostering status; d) final generation source colony; e) final generation end colony.** Significant differences by Wilcoxon Rank Sum Tests were observed between the global number of mutations between strains (W = 28, p < 0.004) and the number of silent mutations between strains (W = 4, p < 0.001). No comparisons between nonsense mutations were made due to representation in only two samples.

strains (1.6e-2 +/- 2.6e-3) compared to NanoLuc strains (8.2e-3 +/- 1.3e-4, W = 0.5, p < 0.001). This appears due to the success of silent mutations (2.3e-2 +/- 4.4e-3, W = 2, p < 0.001) in the F2 generation (3.0e-2 +/- 5.1e-3, W = 0, p < 0.05). No significant differences were found related to the effects of cross-fostering, source colony, or target (end) colony (Fig 5).

*Hierarchical clustering of normalized read counts by sequence position supports strain and generation-specific relationships among samples.* Combining information on mutation location and proportional representation into a dendrogram reveals three major clusters of samples, separating F2 generation. T wild-type samples into two distinct clusters. The non-specific sub-clustering among NanoLuc samples indicates little change in the mutational space but remains distinct from the F0 and F1 wild-type samples within the same cluster (Fig 6).

*Reference-free calculation of Amplicon Sequence Variants (ASVs) reveals strong relationships across dominant quasi-species and overall counts.* Raw counts of ASVs indicate a decrease in F1 followed by an increase in F2 on average for both strains when filtered to a proportional representation >0.1%. This relationship holds true for wild-type samples when filtered >1% representation (S2 Fig). Multisequence Alignment and maximum likelihood tree generation indicate that proportionally dominant quasi-species tend to be closely related, regardless of strain, generation, or cross-fostering status (S3–S5 Fig).

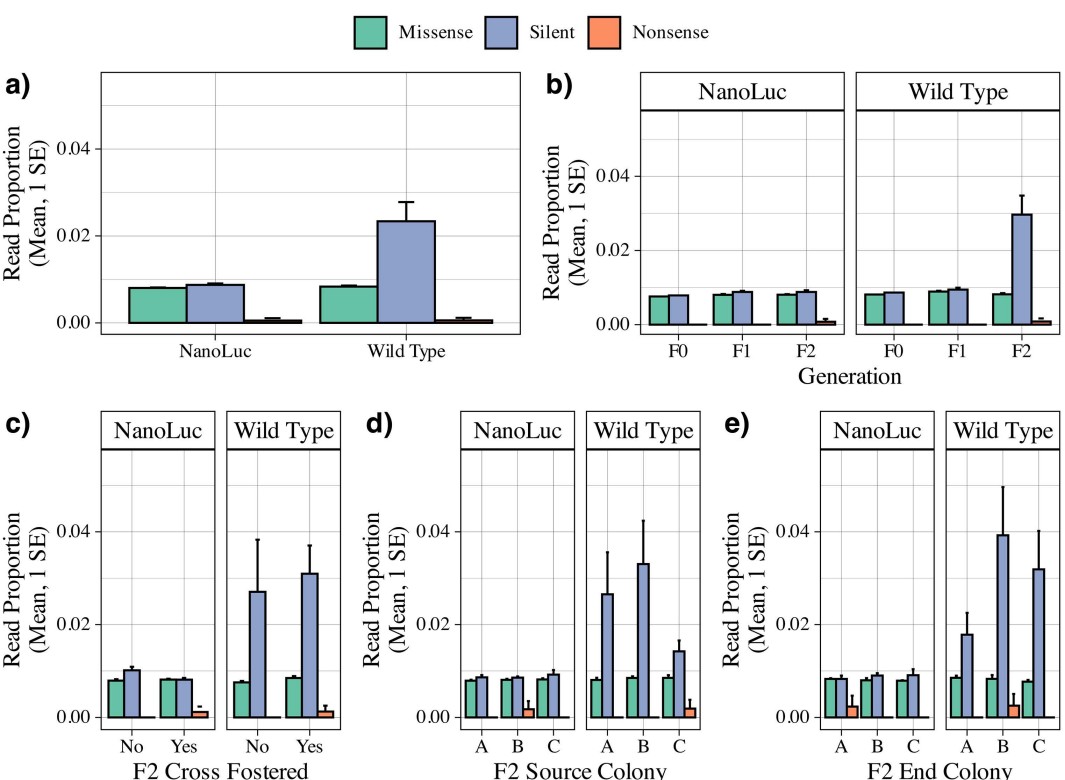

**Fig 5. Normalized proportional representation mean and 1 SE of mutation types across isolated DWV strains a) globally; b) across successive injection generation; c) final generation cross fostering status; d) final generation source colony; e) final generation end colony.** Significant differences by Wilcoxon Rank Sum Tests were observed between the global number of reads with mutations between strains (W = 0.5, p < 0.001), the number of reads with silent mutations between strains (W = 4, p < 0.001), and the number of reads with silent mutations across generations for the wild type strain (W = 0, p < 0.05). No comparisons between nonsense mutations were made due to representation in only two samples.

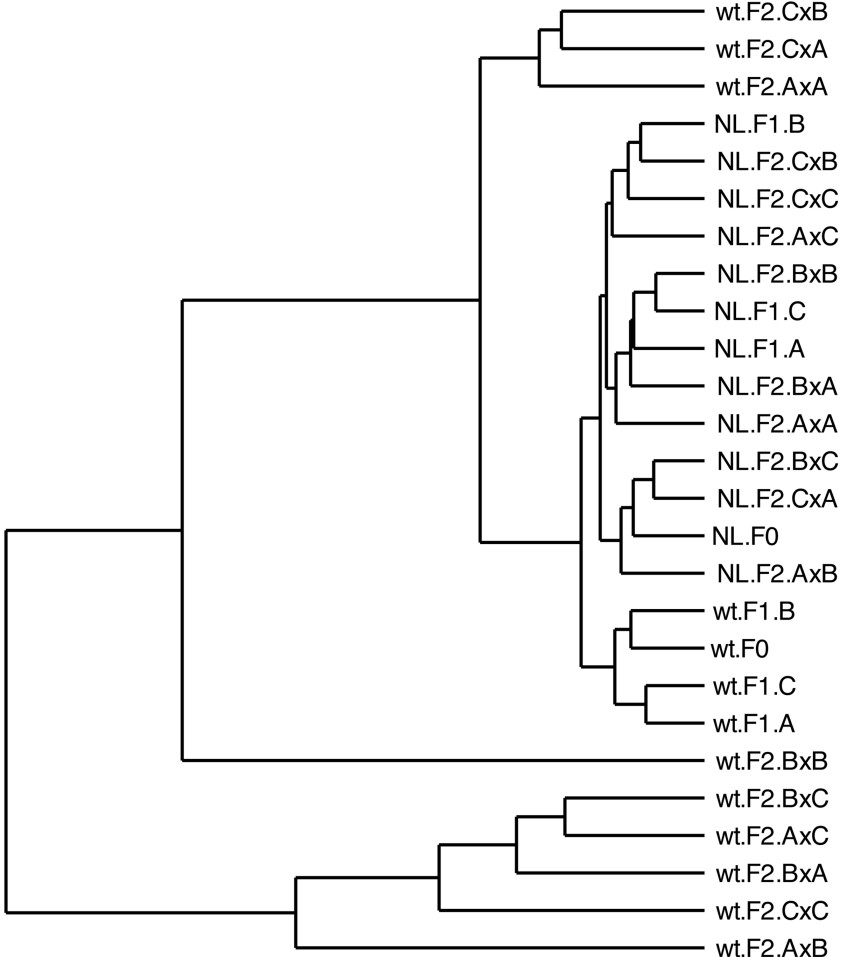

**Fig 6. Hierarchical clustering of normalized mutation read counts per position by sample.** Each sample can be identified by the DWV strain type (NL = NanoLuc, wt = Wild Type), injection generation (F0, F1, F2), and second injection generation source colony (A, B, or C) or third injection generation colony cross-fostering status (denoted by "x").

## Discussion

This study represents a first attempt to explore viral sequence dynamics using a clone of the honey bee virus, Deformed wing virus (DWV). We achieved this by injecting a prepared inoculum through successive generations, either within or across multiple genetic lines, and measuring the mutational effects of nucleotide-level changes for the VP2 structural protein. The results suggest several region-specific mutational hotspots that persist across generations (Fig 3). The type of type, either silent or missense, remained constant within and across generations for the cloned virus. This was true for both the overall number of mutations (Fig 4) and their proportional representation in the read pool (Fig 5). Unsupervised clustering of both mutation location and normalized read count agree that there is no discernable effect observed for injection generation, cross-fostering, or host genetic influence on the mutational space (Fig 6). These results are further confirmed by reference-free quantification and maximum likelihood trees of amplicon sequence variants (S2–S5 Fig).

The presence of wild-type DWV in our samples enabled direct comparison of dynamics between the two strains. Notable is the significant accumulation of silent mutations by the final injection generation of the wild-type strain, in both overall count and proportional representation (Fig 4,5). These effects were also observed through unsupervised clustering (Fig 6), though

the location of accumulated mutations does not appear to differ between strains (Fig 3). In terms of overall number of silent mutations, we did not observe a significant effect of injection generation, but a clear trend of accumulation of silent mutations can be seen between both strands (Fig 4). More injection generations may be required to accurately describe this trend, but the gradual accumulation may help explain the virus' ability to evade immune responses, and RNAi in particular [25].

Significant differences between the NanoLuc and wild-type strains may be explained in part by the 10-fold difference in viral loads (Fig 2). Our qPCR results indicate greater replication rates of wild-type populations, perhaps due to slightly slower replication of NanoLuc clones thanks to added transcription or translation costs [28,29]. Initial development of similar cloned viruses displayed a low-level selection toward insert deletion, though overall stability remained intact at 72 hours post injection [17]. Further, clone-derived DWV Nanoluc [17,18] was based on the DWV variants isolated in 2015 [16]. It is possible that contemporary 2023 DWV variants are better adapted to the bees used in this study compared to 2015. Our qPCR results also reveal a slight decrease in viral loads by the final generation for both strains. This may indicate a decrease in primer efficacy for some quasi-species or an accumulation of non-viable strains in the final inoculum.

That the wild-type strains were present in our samples suggests two possibilities: 1) some level of wild type was present in the initial inoculum or, 2) a latent infection became active when the specimens (bee pupae) used for inoculum preparation were immune challenged [30–32]. The low number of mutations found in the prepared inoculum suggests the specimens were not experiencing an active infection at the time of injection. Also, the small number of closely related quasi species in the inoculum, relative to the final injection generation, supports this thinking (S3 Fig). Furthermore, rapid selection for insertions or deletions is improbable over these time periods, given the known excision rates [18].

The gradual accumulation of silent mutations across generations was unsurprising. The identification of the amino acid changes that lead to successful quasi-species, however, may elucidate further patterns in host-virus interactions. For example, 70% of samples contained a silent mutation at base position 346 due to third-position wobble, with just under half being a cytosine to thymine transversion (S1 Table). This information may prove useful for the development of multi-sequence specific RNAi treatments. And while accounting for all possible combinations across the common mutational positions may be infeasible, perhaps only a threshold is required to supplement the host immune response, providing further detail to the limitations of RNAi. The identification of proportionally dominant genotypes may also aid in such efforts, provided those quasi species remain dominant across more than three generations (S3–S5 Fig).

Altogether this work is focused on mutational patterns in a region of the DWV genome with relatively low nucleotide diversity, we demonstrate that there are specific locations prone to predominantly silent mutations. We also show that the accumulation of mutations may require several successive transmissions to achieve a population of quasi-species capable of evading host immune responses. By using a cloned virus representing a single quasi-species and comparing its mutational progression to the larger population, we gain higher resolution regarding viral dynamics in the host-virus system.

## Methods

### Honey bees

Three experimental honey bee colonies (here labeled A, B, and C) maintained at the USDA-ARS Bee Research Laboratory in Beltsville, MD were used for source specimens. Colonies were regularly inspected and treated as necessary to remain below a 1% *Varroa destructor* infection rate. Brood frames were collected from each colony for differential rounds of infection, and a total of 64 pink-eyed pupae were extracted.

### Injection groups

All pupae were inoculated using a NanoLuc-tagged DWV clone [18] with an automated 20-gauge needle/syringe into the outer abdomen. Injected pupae were incubated at 35°C and 33% humidity for 72 hours and reared using established

methods [16]. Blackened pupae, indicating handling damage or acute infection, were discarded. The injection consisted of a schedule of three successive rounds: first round injections use $10^6$ NanoLuc-tagged DWV clone viral particles as inoculum (F0, 3 pupae). Second-round injections contained DWV isolated from the pooled homogenized solution of round one pupae (F1, 6 pupae per colony). DWV solutions were passed through a 0.2 μm filter to remove cellular material and larger microbes. Third-round injections were prepared in the same way from the second-round homogenate and injected either back into pupae from the same colony or pupae from a different colony (F2, 10 pupae per colony cross) (Fig 1).

## Sequencing

Sample definitions were based on the injection generation (F0, F1, F2), source colony and recipient colony. RNA was extracted from infected pupae using the QIAmp Viral RNA Mini kit and converted using SuperScript Reverse Transcriptase oligo(dT). For each library, viral loads were quantified using qPCR on a BioRad CFX-96 thermal cycler as in [33]), using primers designed for both NanoLuc-tagged DWV and wild-type DWV strains [34]. Nanoluc clones were amplified with the forward primer NanoDWVP1.F (CCTGCTGTTCCGAGTAACCA-3') and the wild-type virus was amplified using DWVP3.F (CCTAATCCAGGACCTGATGGC-3'). Both products were amplified using reverse primer DWVP2.R (TCGCCAATATGC-GTGTACCT −3'). Amplicons were quantified by comparing threshold cycles against honey bee actin (GB44311) amplified in parallel with primers Actin.F (TTGTATGCCAACACTGTCCTTT) and Actin.R (TGGCGCGATGATCTTAATTT). Amplicons for a subset of products were purified and sent to the University of Maryland Institute for Genome Sciences for deep sequencing of isolated cDNA using an Illumina MiSeq. The resulting short-read data represents the DWV structural viral protein VP2 that follows the leader protein. Libraries were sequenced for 1 F0 injection, 3 F1 injections, and 6 F2 injections, for each DWV strain. Sequenced reads are available on NCBI under BioProject ID: PRJNA1295475.

## Data analysis

Raw sequence read data was aligned to a reference sequence derived from the source viral capsid protein with the *bowtie2* aligner [35], using the alignment parameters from *Ryabov et. al. 2019* [16]. All additional data analysis was performed using R programming language 4.3.2 Eye Holes [36]. Sequence alignment map files per sample were used to determine the number of reads representing each mismatched base called at each genomic position. The resulting mismatch counts were normalized by rarefaction to the lowest sample read count. To further obtain accurate and comparable mismatch identification across samples, mismatches with less than a liberal estimate of a 1% sequencing error rate were removed from the analysis after normalization. High confidence mismatches were then translated to determine their codon position, amino acid change, and mutation type (missense, silent, or nonsense) using custom R scripts (Supplemental). The ultimate counts of unique mutations and their normalized read representations were compared using Wilcoxon Rank Sum Tests between inoculum groups and their types of mutations, excluding nonsense mutations due to limited representation across samples. Hierarchical clustering of the mismatch counts and positions per sample was performed using the *hclust* function of the *stats* package as part of base R and plotted using the *ggdendro* package [37]. All additional plots were generated with the *ggplot2* package [38]. Reference-free calculation of amplicon sequence variants and their proportional representations were performed using the *dada2* package [39]. Multisequence alignment was performed with MAFFT v7.526 [40]. The subsequent maximum likelihood trees calculated with RAxML v8 [41], over 1000 bootstrap replicates. The resulting trees were visualized with the *ggtree* [42] and *ggtreeExtra* [43] packages.

## Supporting information

**S1 Fig. Relative transcript abundance (scaled to honey bee transcripts for β-actin) for NanoLuc clone-specific (top) and wild-type primers (bottom) by individual cross of third-round injection cross fostering experiment.** (PDF)

**S2 Fig. Mean number of amplicon sequence variants by strain and injection generation for three levels of proportional representation: > 1%, > 0.1%, and no filter (total).**
(PDF)

**S3 Fig. Maximum likelihood tree generated from multisequence alignment of amplicon sequence variants (ASV) with >1% proportional representation.** Legend further identifies ASVs by their strain, injection generation, cross fostering status, and precise proportional representation.
(PDF)

**S4 Fig. Maximum likelihood tree generated from multisequence alignment of amplicon sequence variants (ASV) with >0.1% proportional representation.** Legend further identifies ASVs by their strain, injection generation, cross fostering status, and precise proportional representation.
(PDF)

**S5 Fig. Maximum likelihood tree generated from multisequence alignment of all amplicon sequence variants (ASV).** Legend further identifies ASVs by their strain, injection generation, cross fostering status, and precise proportional representation.
(PDF)

**S1 Table. Nucleotide changes per sample per base.** Data includes the position, normalized count of mutation at each position, original codon and translated amino acid, new codon and translated amino acid, mutation position within codon, mutation type, proportional read representation, and associated metadata.
(CSV)

**S2 Table. Counts and library sizes for sequenced amplicons by strain, injection generation, source colony, end colony, and cross fostering status.** Counts of unique amplicon sequence variants are broken down into three categories of proportional representation: > 1%, > 0.1%, and no filter (total).
(PDF)

**S1 File. Primers – Forward and reverse primers for both the NanoLuc clone and wild-type DWV strains.**
(DOCX)

**S2 File. Rcode – Custom code written the R programming language that will translate an open reading frame for an existing sequence, then compare it to a data frame of nucleotide polymorphisms at specific locations, and retranslate the amino acid changes into a new data frame.**
(R)

**S3 File. Fasta – Base sequences for both the NanoLuc clone and wild-type DWV strains that were used to compare and derive single nucleotide polymorphisms.**
(FASTA)

## Acknowledgments

We thank Karen Shelton and the Charles Herbert Flowers High School Science and Technology Program for support.

## Author contributions

**Conceptualization:** Eugene Ryabov, Yanping Chen, Jay D. Evans.

**Formal analysis:** Anthony Nearman, Jay D. Evans.

**Investigation:** Alriana Buller-Jarrett, Dawn Boncristiani.

**Methodology:** Anthony Nearman, Eugene Ryabov, Jay D. Evans.

**Writing – original draft:** Anthony Nearman, Jay D. Evans.

**Writing – review & editing:** Eugene Ryabov, Yanping Chen.

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
