## [Decision Letter · Decision Letter 0]

28 Sep 2025

Generational mutation patterns in a honey bee Deformed Wing Virus via infectious clones

PLOS ONE

Dear Dr. Nearman,

Thank you for submitting your manuscript to PLOS ONE. Both reviewers evaluated the manuscript as a valuable contribution that was well presented, and I agree. There are only some minor points raised by reviewer #1 that are worth considering.

We look forward to receiving your revised manuscript.

Kind regards,

Olav Rueppell

Academic Editor

PLOS ONE

2. Please include a separate caption for each figure in your manuscript.

Additional Editor Comments (if provided):

Reviewers' comments:

Reviewer's Responses to Questions

**Comments to the Author**

1. Is the manuscript technically sound, and do the data support the conclusions?

Reviewer #1: Yes

Reviewer #2: Yes

2. Has the statistical analysis been performed appropriately and rigorously?

Reviewer #1: Yes

Reviewer #2: Yes

3. Have the authors made all data underlying the findings in their manuscript fully available?

Reviewer #1: Yes

Reviewer #2: Yes

4. Is the manuscript presented in an intelligible fashion and written in standard English?

Reviewer #1: Yes

Reviewer #2: Yes

Reviewer #1: In “Generational mutation patterns in a honey bee Deformed Wing Virus via infectious

Clones”, Nearman and colleagues explore viral quasispecies dynamics of a critical pathogen of honey bees, Deformed wing virus (DWV). Using previously developed infectious clones, the researchers passaged these viral population through three generations of bees and used qPCR and amplicon sequencing to examine viral load and mutation accumulation through the passaging paradigm. Researchers also infected bees sourced from multiple colonies to measure any possible host effects on virus dynamics. While the infectious clone did not reveal a high amount of variation through passaging, the researchers were able to compare these dynamics to the passaging dynamics of a co-passaged wild type DWV. This wild type DWV demonstrated higher viral titers and a greater rate of silent mutation accumulation through the experiment.

The introduction is a great justification to the study. I only have minor points to improve the manuscript.

Methods :

Why was only VP2 the focus? I know for example full viral genome sequencing incurs higher costs and more computational demand, but justification for this region for these initial experiments would be beneficial.

More details are needed concerning the homogenate prep need to be included for replicability; for example, are pooled homogenates suspended in PBS or some other solution?

L265-272 what specific programs were used for variant filtering and for amino acid translation?

Results/Discussion :

Is it expected that there’s simply more sub-consensus variation in WT vs clones (or detection lower than threshold), which is why you can see the accumulation of silent mutations? This may be addressed by the point made that further passaging beyond F2 may reveal more mutations in the clone, but just to highlight.

In Figure 5, you highlight a higher proportion of reads with mutation by F2, but also F2 viral loads is lower than F1. Is it suspected that accumulation of silent mutations could be interfering with the replication rates of the virus?

I think some additional support into the utility of using these clones for examining virus dynamics would be helpful. The work is centered around the infectious clones (e.g. the title) but it was the lack of variation in the clones compared to the WT virus that emphasized the importance of the WT virus in the passaging paradigm. Or, as authors, are you arguing that these sorts of virus dynamics experiments should instead be conducted with WT virus?

Reviewer #2: The manuscript entitled: "Generational mutation patterns in a honey bee Deformed Wing Virus via infectious clones" exploits the power and novelty of using clones to investigate mutation rates accumulated by viruses in an in vivo setting. The authors focused on a conserved region for obvious applied reasons but it would have been informative if they also contrasted this data set with that obtained from a variable region. That said, the data produced is convincing and therefore has merit in isolation. Congratulations on a really nicely presented paper.

**Do you want your identity to be public for this peer review?** For information about this choice, including consent withdrawal, please see our Privacy Policy

Reviewer #1: No

Reviewer #2: **Yes: ** Declan Schroeder

---

## [Author Response · Author response to Decision Letter 1]

4 Nov 2025

Reviewer #1: In “Generational mutation patterns in a honey bee Deformed Wing Virus via infectious

Clones”, Nearman and colleagues explore viral quasispecies dynamics of a critical pathogen of honey bees, Deformed wing virus (DWV). Using previously developed infectious clones, the researchers passaged these viral population through three generations of bees and used qPCR and amplicon sequencing to examine viral load and mutation accumulation through the passaging paradigm. Researchers also infected bees sourced from multiple colonies to measure any possible host effects on virus dynamics. While the infectious clone did not reveal a high amount of variation through passaging, the researchers were able to compare these dynamics to the passaging dynamics of a co-passaged wild type DWV. This wild type DWV demonstrated higher viral titers and a greater rate of silent mutation accumulation through the experiment.

The introduction is a great justification to the study. I only have minor points to improve the manuscript.

Response: Thank you for your thoughtful review. Your comments certainly helped point out a few shortcomings as well as inspire some deeper thought on the system at hand. We really appreciate it!

Methods :

Why was only VP2 the focus? I know for example full viral genome sequencing incurs higher costs and more computational demand, but justification for this region for these initial experiments would be beneficial.

Response: Great question. VP2 is adjacent to the clone insert, allowing us to design primers that span across the insert into VP2 and select specifically for the clone. We have adjusted the introduction to clarify.

More details are needed concerning the homogenate prep need to be included for replicability; for example, are pooled homogenates suspended in PBS or some other solution?

Response: Certainly! Two previous papers have been published detailing the methods for this system. They are referenced in the introduction on lines 64 and 72, and further throughout the paper.

L265-272 what specific programs were used for variant filtering and for amino acid translation?

Response: Great catch. As the sequence was short, I performed all calculations and sequence translations in R. The variant filtering is detailed in the methods but the translation required custom R code. We will include the code in the final submission and reference it in the methods.

Results/Discussion :

Is it expected that there’s simply more sub-consensus variation in WT vs clones (or detection lower than threshold), which is why you can see the accumulation of silent mutations? This may be addressed by the point made that further passaging beyond F2 may reveal more mutations in the clone, but just to highlight.

Response: This is an interesting thought. It would seem we lack the resolution here to determine exactly when the variant explosion occurred, either at low detection at the end of F1 or the beginning of replication during F2. I suppose we also don’t know what the lag between clone vs wt precisely is, generationally speaking. Certainly, both points can be addressed in future work to better characterize this system. For now, we have provided additional analyses that may help elucidate these points.

In Figure 5, you highlight a higher proportion of reads with mutation by F2, but also F2 viral loads is lower than F1. Is it suspected that accumulation of silent mutations could be interfering with the replication rates of the virus?

Response: Excellent question. We had to think about this for a bit. We landed on two possibilities. One is that sequence diversity may have exceeded the efficacy of our PCR primers by F2 for some quasi species. Another is that there may be some accumulation of non-viable strains in the inoculum. We have updated the discussion with this thinking.

I think some additional support into the utility of using these clones for examining virus dynamics would be helpful. The work is centered around the infectious clones (e.g. the title) but it was the lack of variation in the clones compared to the WT virus that emphasized the importance of the WT virus in the passaging paradigm. Or, as authors, are you arguing that these sorts of virus dynamics experiments should instead be conducted with WT virus?

Response: We are arguing for the use of the cloned species. Given the nature of this particular virus, however, there is always the possibility of latent infection, so it may be that experiments such as these should always be prepared for both. The used of the clone strain allows us to more easily track mutation rates and we expect that, given enough generations, the cloned strain would demonstrate equally high levels of variation. Still, much more work is needed to measure the precise dynamics of this system.

Reviewer #2: The manuscript entitled: "Generational mutation patterns in a honey bee Deformed Wing Virus via infectious clones" exploits the power and novelty of using clones to investigate mutation rates accumulated by viruses in an in vivo setting. The authors focused on a conserved region for obvious applied reasons but it would have been informative if they also contrasted this data set with that obtained from a variable region. That said, the data produced is convincing and therefore has merit in isolation. Congratulations on a really nicely presented paper.

Response: Thank you for the insights and the review! We really appreciate it!

---

## [Editor Report · Decision Letter 1]

6 Nov 2025

Generational mutation patterns in a honey bee Deformed Wing Virus via infectious clones

PONE-D-25-40188R1

Dear Dr. Nearman,

We’re pleased to inform you that your manuscript has been judged scientifically suitable for publication and will be formally accepted for publication once it meets all outstanding technical requirements.

Kind regards,

Olav Rueppell

Academic Editor

PLOS ONE
---

## [Editor Report · Acceptance letter]

PONE-D-25-40188R1

PLOS ONE

Dear Dr. Nearman,

I'm pleased to inform you that your manuscript has been deemed suitable for publication in PLOS ONE. Congratulations! Your manuscript is now being handed over to our production team.

Kind regards,

on behalf of

Dr. Olav Rueppell

Academic Editor

PLOS ONE